# Remote Sensing-Based Land Suitability Analysis for Forest Restoration in Madagascar

**Fitiavana Rajaonarivelo and Roger A. Williams ***

School of Environment and Natural Resources, The Ohio State University, Columbus, OH 43210, USA
* Correspondence: williams.1577@osu.edu; Tel.: +1-614-688-4061

**Abstract:** Tropical forest loss has been a prominent concern in Madagascar, portrayed by a highly fragmented landscape of forests surrounded by small-scale agricultural patches along the northwestern and eastern regions. This paper seeks to identify priority areas for forest landscape restoration at a landscape scale using a geospatial land suitability analysis approach. The study area is the watershed of Mahavavy, home to one of the most biodiverse ecosystems in the island but also an important agriculture region. The suitability analysis method comprises five major steps: (1) the identification of a set of restoration criteria, (2) the acquisition of available environmental datasets for each criterion, (3) generating suitability maps for each criterion, (4) the conception of a suitability analysis model, and (5) the design of potential sites for restoration. The specific suitability criteria combine both landscape composition and soil characteristics, including (i) distance from protected sites and forest patches, (ii) land cover classes, (iii) distance from settlements, (iv) distance from roads, (v) risk of soil erosion. We found 143,680 ha (27.9%) that were highly suitable areas for forest landscape restoration, 159127 (30.9%) moderately suitable and138031 ha (26.6%) not suitable areas. High potential suitable areas are observed in close proximity of forest patches and protected areas, and low restoration feasibility in all areas that are easily accessible and thus subjected to exploitation.

**Keywords:** forest restoration; landscape restoration; remote sensing; land suitability; Madagascar; fragmented landscape; GIS





## 1. Introduction

In the tropics, the increasing loss of forest areas and biodiversity due predominantly to agricultural land expansion has raised concern towards the end of the twentieth century [1–5]. In spite of a surge in forest protection and management efforts, land use intensification trend has been amplified in several places during the past decades, mostly in South East Asia and East Africa [5–7]. Madagascar, an island located off the Southeast coast of Africa, is not exempt to this trend and has been a prominent concern regarding tropical forest loss [8–11]. The country is one of the highest priority areas for conservation in the world [12–14] due to its exceptional biodiversity and high levels of endemism in terms of species and several taxonomic groups [15–17], land use expansion has severely threatened this rich biodiversity, leaving a highly fragmented landscape of forests, surrounded by small-scale agricultural patches along the northwestern and eastern regions of the country [7,11,18–20].

However, addressing those most important challenges threatening biodiversity management and conservation in the 21st century cannot be limited exclusively on conservation [21,22]. For this reason, a landscape-based restoration, also known as "Forest Landscape Restoration (FLR)", has been actively proposed and widely applied in diverse ways as a framework to complement forest and biodiversity conservation worldwide [23–25]. The term FLR was described as "a planned process that aims to regain ecological integrity and enhance human wellbeing in deforested or degraded landscapes" [21,25,26], in which the primary goal is not to increase forest cover or restore the ecosystem to its original

state, but rather to create a forest-based landscape that is beneficial to both nature and humans [24,27–29]. Despite the fact that various research and reports on FLR have been published [23], there are few scientific papers that explain which tools to utilize in a practical implementation [23–25].

What makes the complexity of applying FLR in practice is its multi-objective nature and landscape-based planning, one of the key issues in selecting restoration priority areas [28]. Prioritization of forest restoration areas is a common issue in nature conservation projects given the expectation of satisfying a diverse set of socioeconomic and ecological objectives [30]. Further, quantitative decision support tools are required to help examine landscape spatial patterns in order to select priority areas sites, to add different categories of variables given the multi-objective nature of the problem, and evaluate tradeoffs between different restoration techniques [28,30]. Previous studies have developed methodologies based on Multicriteria Analysis (MCA) and Geographical Information Systems (GIS) that have proven to be successful in a variety of applications related to forest conservation, management, and restoration, and environmental planning [28,31–38].

Multicriteria Analysis offers a diverse set of tools and techniques for organizing decision problems as well as constructing, analyzing, and ranking alternative options [33,39]. It provides a framework for including many objectives and assessment criteria, most of which are in conflict, weighting them based on their relevance, and selecting the most appropriate outcomes [32,33]. In addition to the multi-objective character of FLR, GIS tool is also required to conveniently manage spatial data [28]. For this purpose, the prioritization using MCA and GIS offers a robust framework and tool to perform a suitability assessment of forest restoration sites.

This study presents spatially explicit method based on land suitability assessment for the identification of restoration priority areas at landscape level to understand the feasibility of restoration in Madagascar. We chose the watershed of Mahavavy, located in the northwestern region, as a case study due to its importance both in forest biodiversity and agricultural activities. In the context of watershed protection and biodiversity conservation, restoration objectives are particularly related to increasing vegetation and forest cover in degraded land to prevent and control erosion and protect forest habitats [8–10]. To achieve this goal, the specific objectives will focus on (i) creating suitability maps for each criteria identified, and (ii) designing options for priority restoration areas. Thus, this paper will support the regional and national studies conducted so far on FLR in Madagascar, given the commitment made to the United Nations Framework Convention of Climate Change to reforest 270,000 ha with native species [40] and the Bonn Challenge to restore 4 million hectares of degraded forests by 2030 [41].

## 2. Materials and Methods

### 2.1. The Study Area

The study area is the watershed of Mahavavy located in the north-western part of Madagascar (Figure 1), which is home to various extraordinary ecosystems and biodiversity for which Madagascar is known. We chose the watershed of Mahavavy, located in the northwestern region, as a case study due to its importance both in forest biodiversity and agricultural activities. The watershed covers an area of 147 km$^2$ with an elevation that varies between 0 to around 2850 m a.s.l.

The precipitation and temperatures that are observed are typical of subtropical climate, with an austral summer from December to April (rainy season) and an austral winter from May to November (dry season) [42]. Throughout the year, the average temperature is 27 °C, ranging from 18 °C to 34 °C and is rarely less than 16 °C or above 36 °C. Precipitation averages 1280.1 mm and most humid season lasts about 5 months, from December to April with a daily precipitation probability greater than 38%.

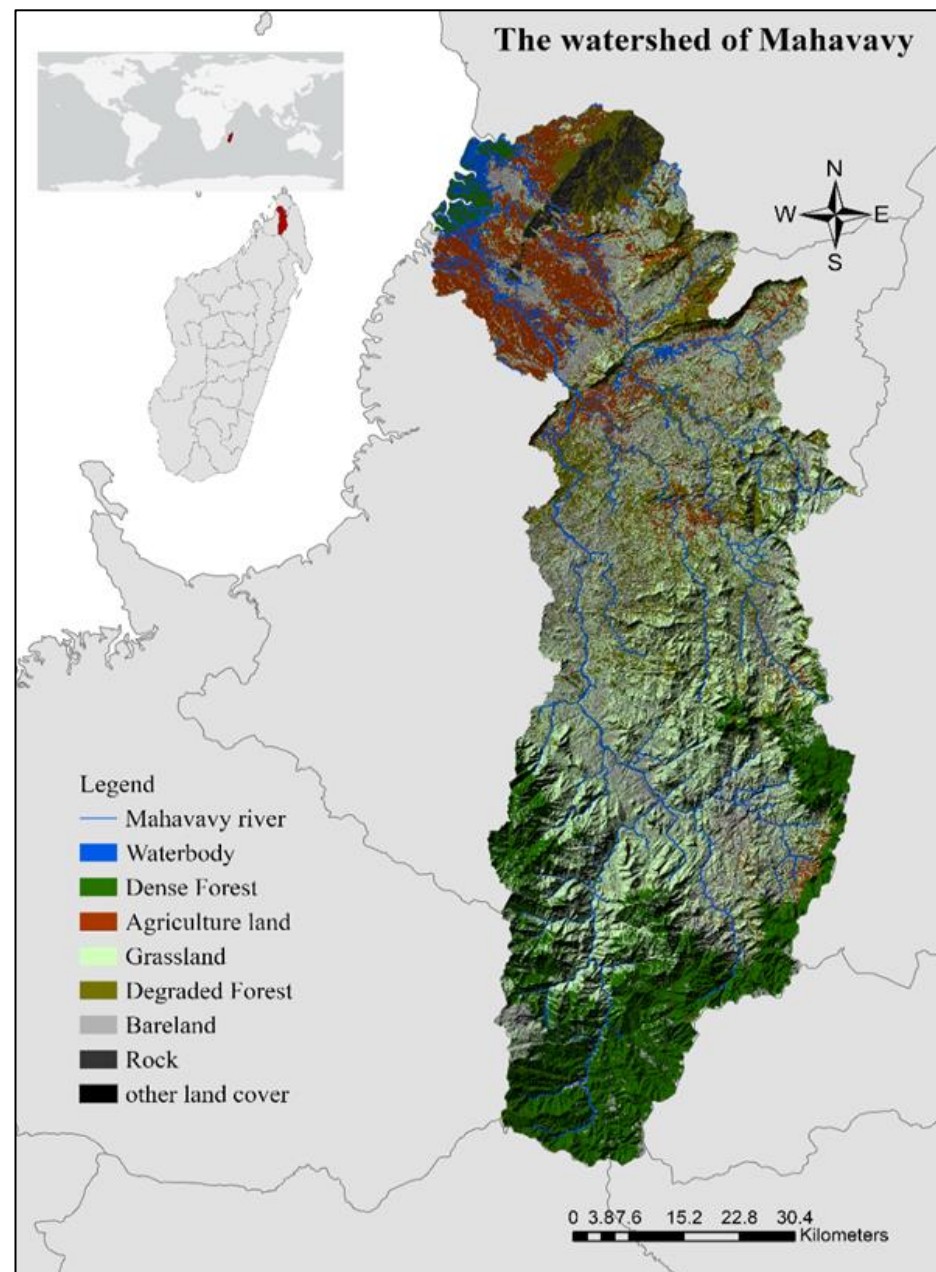

**Figure 1.** The Mahavavy watershed in northwestern Madagascar.

The region is composed of relics of mangroves, along the coastal areas, and primary forest of dense, low-canopy, seasonally moist, semideciduous secondary forest [15]. The remaining dense forests are mostly observed in protected areas in the mountainous part of the region [11], which are the main source of the river of Mahavavy. Down to the lowland, the landscapes of the watershed can be described as a mosaic of small fragments of degraded forests surrounded by small plots with diverse land uses dominated by rice fields and commercial crops such as sugar cane and cashew. Traditional shifting cultivation is used to produce rain-fed upland rice on moderate to steep slopes. The common practice is to clear and burn small plots, after which rice is planted [43].

### 2.2. The Suitability Analysis

Recently, land suitability analysis is considered as an important step for land-use management, planning, and restoration [43–46]. The objective of land suitability analysis is to identify the potential suitability of a particular area for specific land-use based on a wide

range of criteria in terms of environment, social, and economic factors [28,44]. Therefore, in this paper, the suitable areas for forest restoration refers to the lands that provide the maximum benefits for ecological restoration with socio-economic advantages over a period of time [6].

The suitability analysis is based on:

(1) identification of a set of restoration criteria
(2) acquisition of available GIS datasets for each criterion
(3) generating suitability maps for each criterion
(4) conception of a suitability analysis model
(5) design of potential sites for restoration.

2.2.1. Identification of Criteria for Land Suitability Analysis

To perform the land suitability analysis, Orsi and Geneletti [28] in their study of identification of priority areas for FLR, propose to consider multiple criteria that can be grouped in two objectives: ecological and socioeconomic. The basic assumption of the method is captured in the following equation [28]:

$$\text{Restoration priority} = f(E, SE) \tag{1}$$

The Ecological factor represents the necessity for restoration to protect ecologically important sites and increase forest cover, while the socio-economic factor represents the feasibility of the restoration activity depending on the socio-economic context of the region.

The selection of criteria to assess E and ES factors of Equation (1) was driven by literature review and the need to be spatially represented. Furthermore, the availability of georeferenced data for the region was considered in choosing the criteria.

The following criteria were selected for ecological factor (E):

1. Land cover class- The land cover classes of interest include seven categories which are: dense forest (primary forest and mangroves), degraded forest (degraded continuous forest and forest fragments), grassland (nonwoody vegetation such as grasses, herbaceous plants), agriculture areas (mostly rice field, commercial crops, tannes. The term 'tannes' refers to the inner part of mangroves, coastal wetlands found in tropical and subtropical regions. They represent the least frequently submerged zones, with soils generally oversalted or acidified, developing at the expense of a mangrove. We distinguish between "naked tannes" and "herbaceous tannes" according to the vegetation cover in mangrove areas, agroforestry), and minor land cover classes such as waterbody (river, lake, stream), bare land, and limestone massif (classified as rock). Forests (dense and degraded) and grassland were given priority in our criteria to assess the Ecological factor (Figure 2a).

2. Distance from protected sites and forest patches—Restoration in and around a protected site and forest patches means both enhancing the forested ecosystem and creating a buffer zone that prevents the site from being disturbed. In addition, areas around existing forests are a priority for their proximity to reservoirs of native species [28]. The location of protected sites and forest cover in the watershed region are shown in Figure 2b.

The following criteria were selected for socioeconomic factor (SE):

1. Settlements—Settlements include human habitation and build infrastructures (hospitals, schools, churches and sports areas). High concentration of human activities in cities and villages are considered a driver for land degradation and natural resources exploitation in the nearby forest area. In a study of deforestation in the northeast part of the country [11], it was proposed that the range of influence of land users is at a maximum of about 2.5 km from the home village, which can be extrapolated to about 20 km² neighborhood area from the village. A settlement map is produced using Landsat 8 OLI satellite images and verified with Google Earth map (Figure 2c).

2. Distance from a road—Although roads facilitate restoration activities, they are mostly considered a source of disturbance because they increase the access of nearby forest area et expose them to exploitation [6,28]. The existing roads network in the watershed region is shown in Figure 2d.

3. Risk of soil erosion—To achieve restoration objectives in a watershed, soil loss due to erosion is an important criterion to consider in choosing priority areas for restoration. Preventing and stabilizing erosion-prone land by increasing the vegetation cover is among the main objectives of restoration activities in a watershed [28]. A soil erosion map is produced using a method called Revised Universal Soil Loss Equation (RUSLE), explained in Section 2.2.3—Soil Erosion Map.

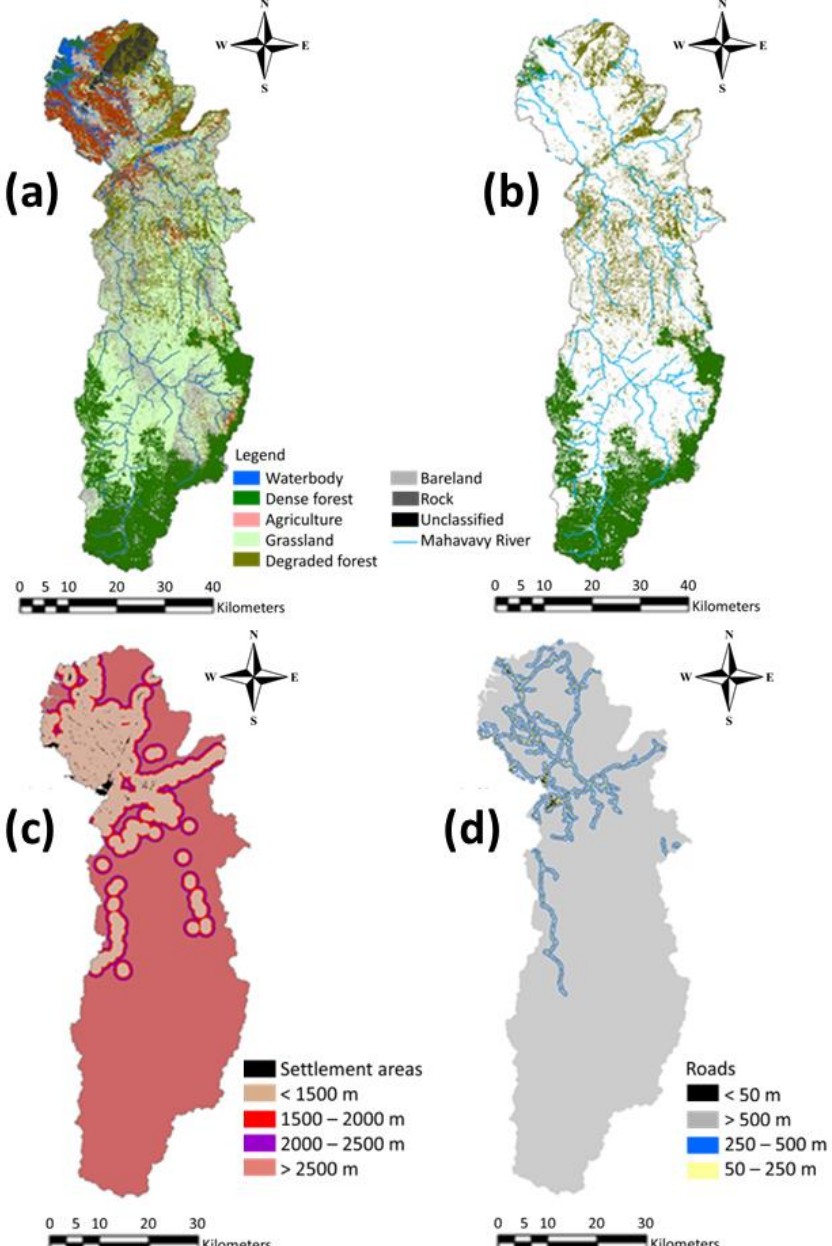

**Figure 2.** Criteria for the land suitability analysis for priority restoration areas. (**a**) LULC, (**b**) Forest cover, (**c**) distance from settlements, (**d**) distance from roads.

2.2.2. Data Acquisition and Sources

Land use/Land cover (LULC) map was generated from multispectral datasets of 2021 of the study area using Landsat 8 OLI. Data were collected from the USGS satellite remote

sensing datasets resources. Other datasets for each selected criteria were collected from open-source websites (Table 1).

**Table 1.** Satellite and GIS datasets for land suitability analysis for extension of potential forest areas in northwest Madagascar [6,7,28].

| No. | Criteria | Scale/Resolution | Data Source |
|---|---|---|---|
| 1 | Land use/Land cover | 30 m resolution | 2021, Landsat 8 OLI |
| 2 | Ambilobe District boundary Mahavavy Watershed boundary | 1:13,300,000 | Dataset- Humanitarian Data Exchange https://data.humdata.org. (Last accessed 19 October 2022) |
| 3 | Settlement | 30 m resolution | Landsat 8, Google Earth |
| 4 | Roads | 1:1,300,000 | Madagascar—Geofabrik Download Server https://download.geofabrik.de/africa/madagascar.html (Last accessed 19 October 2022 ) |
| 5 | Soil erosion | 1:1,300,000 | Elevation data using ASTER Global Digital Elevation Model (GDEM) from Earthdata.nasa.gov Global Soil Data from FAO Database https://data.apps.fao.org/map/catalog/srv/eng/catalog.search#/metadata/446ed430-8383-11db-b9b2-000d939bc5d8 (last accessed 19 October 2022) Rainfall Data 1901–2021 from CRU Database https://crudata.uea.ac.uk/cru/data/hrg/ (Last accessed 19 October 2022) |

### 2.2.3. Map Generation for Each Criterion

A map was generated for each criterion with a 30-m cell size using GIS operations. Map for boundary and roads were already available for the study area through open-source database (Table 1). Map for settlements was processed using Landsat 8 and Google Earth images. A classification map from Landsat 8 images was generated, using a maximum likelihood supervised classification to locate the possible areas classified as settlements. A digitalization of those areas was then processed using Google Earth images as a reference to add more precision to the map.

Maps of LULC and soil erosion also need to be computed, as explained in the following paragraphs.

#### Image Processing for Land Use/Land Cover (LULC)

Satellite datasets from Landsat 8 OLI (2021) were used to develop the LULC image processing from the USGS satellite remote sensing datasets resources. To prepare the classification maps, the satellite images were visually interpreted using a maximum likelihood supervised classification in the ArcMap environment, using Google Earth images as a reference. The maximum-likelihood classifier was adopted from a parametric classification algorithm and divided into seven classes: dense forest, degraded forest, grassland, agricultural land, waterbody, bare land, and rock. The classification process is based on the Land cover classification system (LCCS) developed by the Food and Agriculture Organization of the United Nations (FAO) for Africa [47–49].

#### Accuracy Assessment for LULC Classification Map

The accuracy assessment process consists of verifying the reliability of the previously produced LULC classification map. The assessment includes estimating the accuracy of the LULC classification by establishing random sample points on the classification map and comparing the same points on a reference map such as Google Earth image for 2021 to see if the category is identical. The comparison is conducted using a confusion matrix which represents the accuracy of the map, and also allows to determine the categories which are

likely to be easily confused with others. We selected 210 random points on the map. The accuracy was estimated using the Producer's Accuracy (PA), User's Accuracy (UA) and Overall Accuracy (OA) [6,7].

The PA represents the percentage of the real data on the ground that are classified correctly on the classification map, while the UA indicates the proportion that a data classified into a given category on the map is represented correctly on the ground. The UA also considers classification errors on the map to allow a better assessment of the quality of the classification.

The OA can be evaluated either from an average of PA and UA or from the Kappa coefficient, a statistic that measures the agreement, out of probability, between the two maps (Classification Map and Ground Reference Map) [50].

Soil Erosion Map

Soil erosion factor can be obtained from soil loss assessment using simple method called Revised Universal Soil Loss Equation (RUSLE) [51,52], summarized in the following equation:

$$A = R \times K \times (LS) \times C \times P \tag{2}$$

where:

A = the average annual soil loss (ton ha$^{-1}$ yr$^{-1}$)
R = the rainfall-runoff erosivity factor (MJ mm ha$^{-1}$ hr$^{-1}$ yr$^{-1}$)
K = soil erodibility factor (ton ha hr MJ$^{-1}$ ha$^{-1}$ mm$^{-1}$)
LS = topographic steepness factor (based on length and slope)
C = land cover management factor
P = erosion-control practices factor

The amount of soil loss is strongly dependent on rainfall, runoff, type of soil, vegetation, biotic/abiotic disturbances, and topographic characteristics [51–54]. Each factor (R, K, LS, C, P) was processed and quantified using different equations from literature review (Table 2) and computed in ArcGIS environment to generate a map for each one (Figure 3).

**Table 2.** Description of each factor used in RUSLE equation for estimating soil loss.

| Factor | Estimation of Factor | Description | Source |
|---|---|---|---|
| R | $$R = 0.5 \times IDWI \times 1.73 \tag{3}$$ where: <br> • IDWI is the extrapolation of rainfall point data to the study area | It quantifies the amount of runoff associated with the rain and was computed using global rainfall information from the Climatic Research Unit Database, based on 10-year average rainfall record (2011–2021). | [55] |
| K | $$K = Fcsand \times Fcl.si \times Forgc \times Fhisand \tag{4}$$ where: <br> • Fcsand is a factor that gives low soil erodibility for soils with high coarse-sand contents and high values for soils with little sand <br> • Fcl.si gives low soil erodibility factors for soils with high clay-to-silt ratios <br> • Forgc reduces K values in soils with high organic carbon content, <br> • Fhisand lowers K values for soils with extremely high sand content | It represents the susceptibility of soil to erosion depending on soil properties, rainfall, and runoff. Global Soil Data from the FAO Database was used to process K factor. | [56] |

<div align="center">**Table 2.** *Cont.*</div>

| Factor | Estimation of Factor | Description | Source |
|---|---|---|---|
| LS | $LS = \left( \frac{\text{Flow accumulation} * \text{Cell size}}{22.13} \right)^{0.4} * \left( \frac{\text{Sin(slope)}}{0.0896} \right)^{1.3}$ (5) <br> where: <br> • Flow accumulation is the total amount of upslope contribution for a given cell <br> • cell size is the size of grid cell (30 m for our case) <br> • *sin* (*slope*) is the sinus value of slope. | It reflects the influence of topography on soil loss and computed using elevation data. | [57] |
| C, P | Refer to Table 3 | C factor denotes the rate of soil loss associated with a land use class and management practices and calculated using land cover classification of the study area from the classified satellite data. <br> P factor reflects the rate of soil loss associated with support conservation practices such as contour farming, strip cropping, and terracing. | [52,58] |

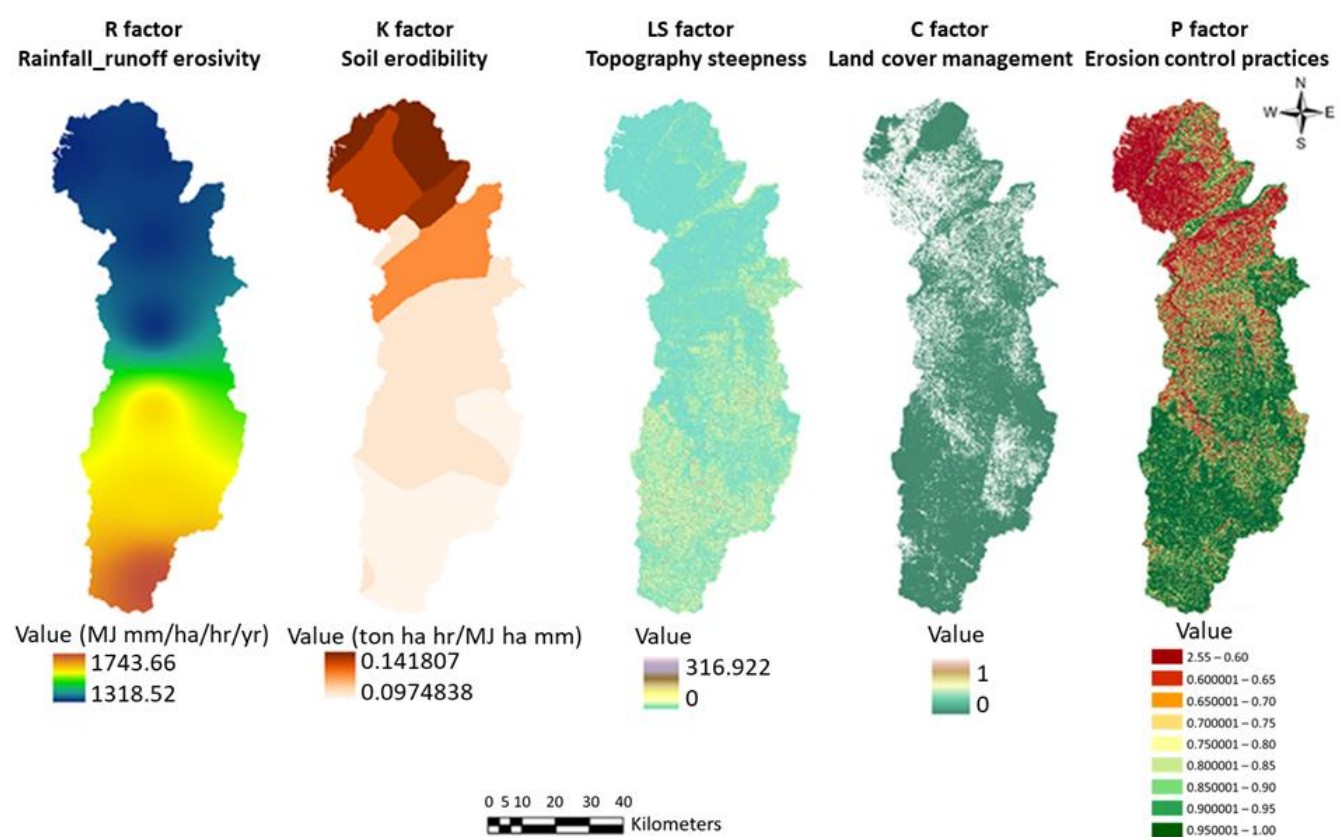

**Figure 3.** Key inputs variables for (R, K, LS, C, P) soil loss assessment.

**Table 3.** The C factor for different land use land cover (LULC) classes and the P factor according to the slope (data source from [56,59]).

| The C Factor | | The P Factor | |
|---|---|---|---|
| Land Use/Land Cover Class | C Factor | Slope (%) | Contouring |
| Dense forest | 0.0015 | 0.0–7.0 | 0.55 |
| Degraded forest | 0.0200 | 7.0–11.3 | 0.60 |
| Agriculture land | 0.4000 | 11.3–17.6 | 0.80 |
| Water | 0.0000 | 17.6–26.8 | 0.90 |
| Rock | 0.0000 | >26.8 | 1.00 |
| Grassland | 0.0150 | | |

As the limitation of this study, the estimation of soil loss quantification and mapping as well as the comparison of the erosion values were based on the soil loss potential of the Abay river basin in Ethiopia, Africa [60] due to the lack of similar studies in the study region.

### 2.2.4. Conception of the Land Suitability Analysis Model

The following framework summarize the process to generate a land suitability analysis (Figure 4).

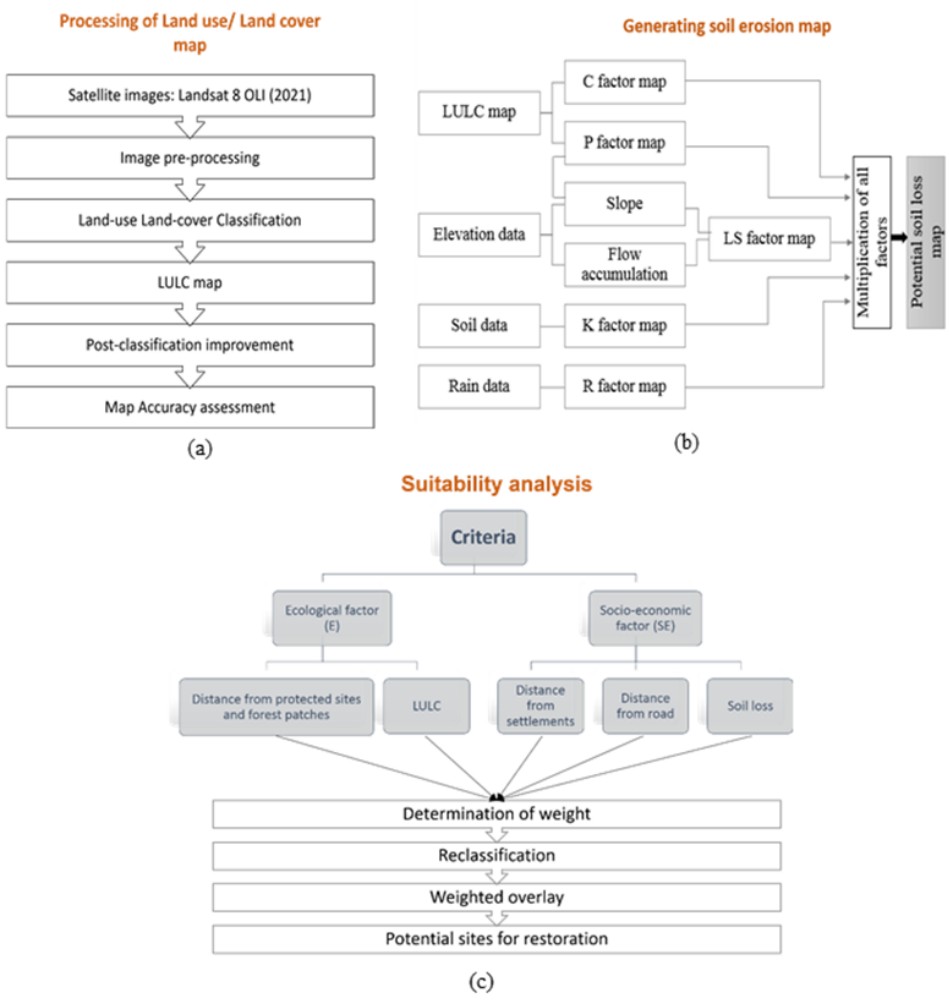

**Figure 4.** Research framework for land suitability analysis to propose the priority areas for landscape restoration (**a**) Land use/Land cover classification processing. (**b**) Soil erosion map processing, (**c**) Land suitability analysis.

- Reclassification of each criterion map: Once the five maps are generated, they are combined using a multicriteria analysis to generate the final suitability map. However, to make them comparable before the computation, a reclassification process was performed. Reclassification of the selected criteria is performed to better understand the importance of each criterion in the suitability assessment. It consists of replacing a new value in each criterion map to reclassify the vector and raster data [6,28]. The reclassification is based on the following levels of suitability: highly suitable (S1), moderately suitable (S2), marginally suitable (S3) and not suitable (N) (Table 4).
- Distance from protected areas and forest patches: The ranking system was based on natural forest regeneration through seed dispersal limitations Natural regeneration is highly favored in areas within an 100 m radius via short-distance dispersal (e.g., wind, gravity) [61,62] and still be possible via longer-distance dispersal within 100–1000 m (e.g., birds, bats, primates) [62].
- Soil loss priority and severity ranking: This was based on soil loss potential of the Abay river basin in Ethiopia, Africa, in which catastrophic soil loss is classified as greater than 500 t ha$^{-1}$ yr$^{-1}$, severe is between 50 to 500 t ha$^{-1}$ yr$^{-1}$, and less than 50 t ha$^{-1}$ yr$^{-1}$ is moderate to slight soil loss [61].

**Table 4.** Reclassification of the criteria for land suitability analysis (data source from: [1,23]).

| Criteria | Suitability Class | | | | Literature |
|---|---|---|---|---|---|
| | Highly Suitable (S1) | Moderately Suitable (S2) | Marginally Suitable (S3) | Not Suitable (N) | |
| Distance from protected areas and forest patches (m) | <100 | 100–500 | 500–1000 | >1000 | [61,62] |
| LULC | Forest (dense and degraded) | Grassland | Agriculture land, Bare land | Water, Rock | [6] |
| Soil loss (t ha$^{-1}$ yr$^{-1}$) | >500 | 50–500 | 0–50 | | [63] |
| Distance from settlements (m) | >2500 | 2000–2500 | 1500–2000 | <1500 | [7,11] |
| Distance from roads (m) | >500 | 250–500 | 50–250 | <50 | [63] |

- Distance from settlements: we based the rank on the assessment made by [11] in north-eastern Madagascar, in which they conclude the range of influence of land users within a radius of about 2.5 km from the home village. We therefore set the distance of the most suitable areas to greater than 2500 m from settlements. However, we considered, at our own discretion, the radius within 1500–2500 m as moderately and marginally suitable because of some restoration activities such as agroforestry that local people usually practice around their villages [7,11].
- Distance from roads: the suitability class for distance from roads was estimated from a buffer analysis, based on previous urban development studies [64], and the influence of human disturbance from road infrastructures. Based on buffer analysis, the spatial agglomeration impacts of road infrastructures in the region are within 500 m in large cities and within 50 m in rural areas. Therefore, we set a margin value between 50 to 500 m as the basic buffer value.
- Designing Options for Priority Restoration Sites: The basic assumption that guided the process of land suitability is that a site can be restored only if it is sufficiently within or around forests and nature reserves; not in close proximity to agriculture, roads and settlements; and in erosion-prone land areas [28]. In other words, the most suitable site to be restored is that minimizing landscape fragmentation (distance from protected areas and forest patches, LULC), minimizing conflicts with human settlements, infrastructures, and agricultural fields (distance from settlements and roads, LULC), minimize soil erosion (soil loss). The rationale for this assumption was that restoring erosion-prone lands within a watershed with potential agricultural

outcomes is likely to improve provision of ecosystem services, in particular water and soil quality, precisely where people are most exposed to land degradation, as in many rural areas in the Mahavavy region. For this reason, criteria related to distance from protected areas and forest patches, LULC, and soil loss were considered benefits, while criteria related to distance from settlements and distance from roads were considered costs.

Finally, weights were assigned to each criterion (forest patches, LULC, settlements, roads, soil erosion) to compute the final suitability map for priority restoration areas [6,28]. Weights were assigned equally to the groups ecological and socioeconomic (0.5: 0.5) [28], in which produces higher weight values to criteria in the ecological group (Table 5)

**Table 5.** Weight assigned to each group and criteria in assessing the suitability of land in the watershed of Mahavavy.

| Weight | | | | |
|---|---|---|---|---|
| **Ecological Factors** | | | **Socio-Economic Factors** | |
| Distance from protected areas and forest patches | LULC | Soil loss | Distance from settlements | Distance from roads |
| 0.25 | 0.25 | 0.16 | 0.17 | 0.17 |
| Total weight: 0.50 | | | Total weight: 0.50 | |

## 3. Results

### 3.1. Soil Quantification and Mapping

The results of the RUSLE-GIS approach are used to assess the spatial distribution of soil erosion, identify vulnerable areas and determine average soil losses, in t ha$^{-1}$ yr$^{-1}$. Total soil losses in the Mahavavy watershed range from 0 to 59,059 t/ha/yr. Figure 5 shows that areas with severe risk of erosion (greater than 500 t ha$^{-1}$ yr$^{-1}$) are mostly located along the bank of the streams in the mountainous part of the regions. Those areas are located on hills and steep slopes and account for 7.53% of the watershed (384.14 ha). Areas of low to moderate erosion sensitivity (between 50–500 t ha$^{-1}$ yr$^{-1}$) are mostly located in the southern and eastern part of the watershed, and account for 11.93% (6082.94 ha). Areas with insignificant risk of erosion account (less than 50 t ha$^{-1}$ yr$^{-1}$) for 80.54% (503,495.60 ha).

### 3.2. Accuracy Assessment for LULC Classification

The overall accuracy of the LULC classification is 87.1% and the Kappa coefficient is 0.85 (Table 6), which can be interpreted as a high level of quality for a classification map [7]. The producer's accuracy is high for the dense forest (91.67%), agriculture land (93.54%), waterbody (100%), bare land (100%), and rock (100%) classes, but lower for the degraded forest (76.19%) and grassland (72.09%) classes.

Those PA values mean that error in classification for dense forest, agriculture land, waterbody, bare land, and rock are low. For the user's accuracy, only the bare land class has a low percentage value (71.43%), because it can be confused with urban areas, or dry stream.

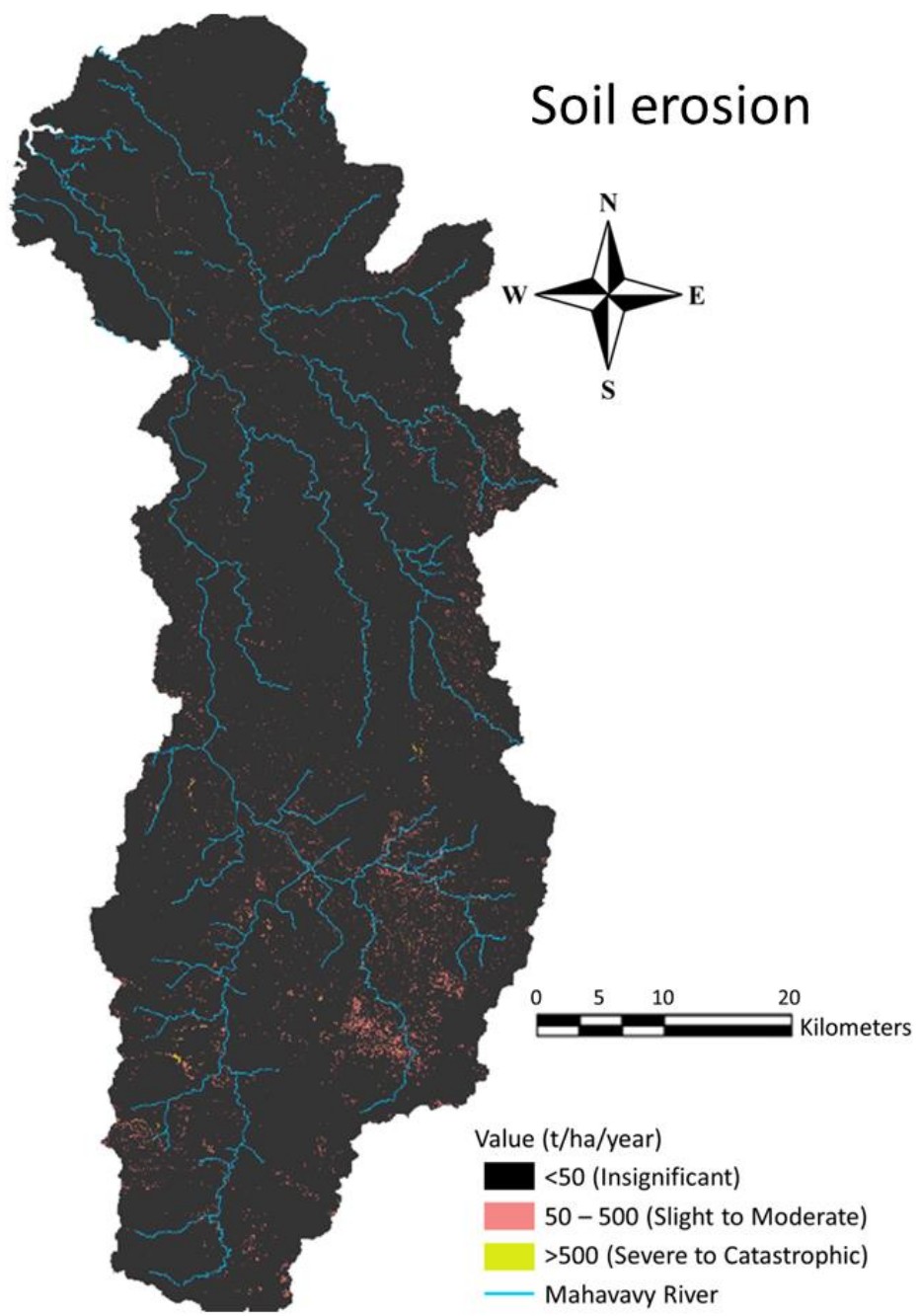

**Figure 5.** Soil erosion map of the watershed of Mahavavy.

**Table 6.** Accuracy assessments for Land use/Land cover (LULC) classification.

| LULC Classification | Accuracy Assessment (%) | | | |
|---|---|---|---|---|
| | Producer's Accuracy | User's Accuracy | Overall Accuracy | Kappa Coefficient |
| Dense forest | 91.67 | 94.28 | | |
| Degraded forest | 76.19 | 91.42 | | |
| Grassland | 72.09 | 88.57 | | |
| Agriculture land | 93.54 | 85.29 | 87.1 | 0.85 |
| Waterbody | 100 | 96 | | |
| Bare land | 100 | 71.43 | | |
| Rock | 100 | 81.81 | | |

*3.3. Land Suitability Analysis for Priority Restoration Areas*

3.3.1. Reclassification of Criteria

Each criteria map is classified into four categories: highly suitable (S1), moderately suitable (S2), marginally suitable (S3) and not suitable (N) [6]. The distance from protected areas and forest patches criteria shows that 58.14% (299,311.79 ha) of the areas are highly suitable, and 32.29% (166,239.45 ha) moderately suitable (Figures 6 and 7a). For the LULC map, 32.65% (168,024.62 ha) of the areas are covered by forest (dense and degraded), which are considered highly suitable, and 34.60% (178,093.06 ha) are covered by grassland, which is considered moderately suitable (Figures 6 and 7b). Related to risk of soil erosion, areas that should be given priority for restoration accounts for 0.08% (381.13 ha) and 1.19% (6082 ha) for highly and moderately suitable, respectively (Figures 6 and 7e).

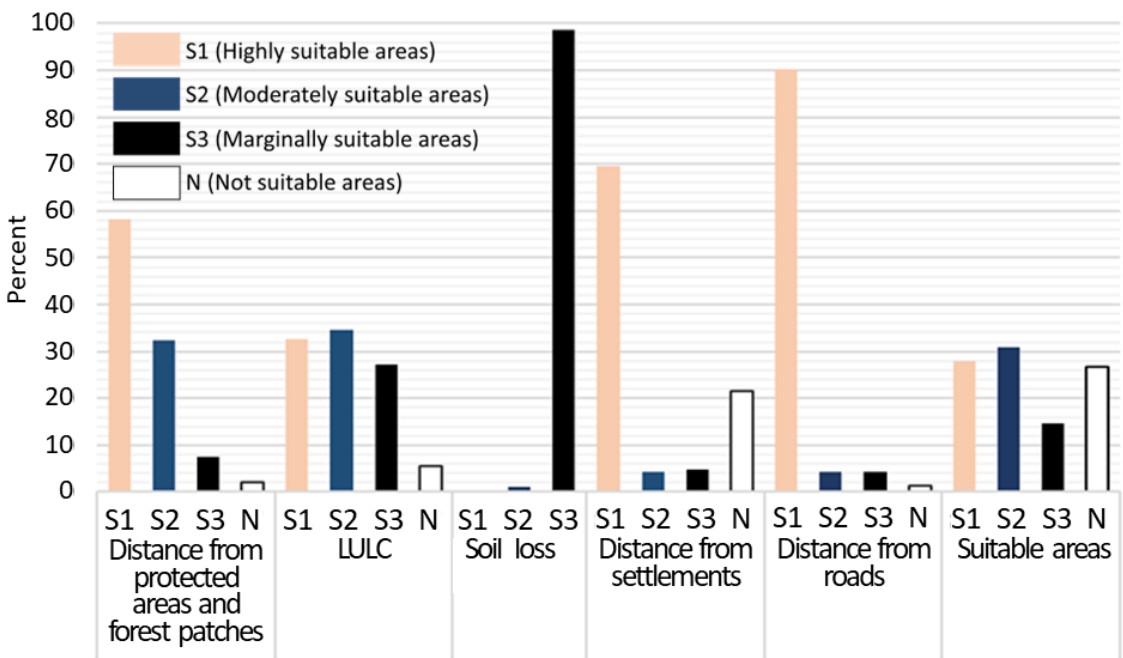

**Figure 6.** Reclassification of the criteria for land suitability analysis and percentage area for each class.

By considering the distance from settlements, 69.52% (357,889.27 ha) of the areas are highly suitable and 4.2% (21,595.06 ha) are moderately suitable (Figures 6 and 7c), while from roads, 90.18% (464,224.13 ha) of the areas are highly suitable and 4.28% (6825.43 ha) are moderately suitable (Figures 6 and 7d).

3.3.2. Land Suitability Analysis

To select the potential areas for restoration in the Mahavavy watershed, five criteria were used, as shown in Figure 7a–e. The highest preferences are the highly suitable class (S1) followed by the moderately suitable class (S2). With a total area of 514,789.2 ha, there is possibility of forest restoration in 27.91% (143,680.14 ha) in highly suitable areas and 30.91% (159,127.63 ha) in moderately suitable areas (Figure 7f). Not suitable areas in the watershed accounts for 26.62% (137,031.23 ha) of the areas.

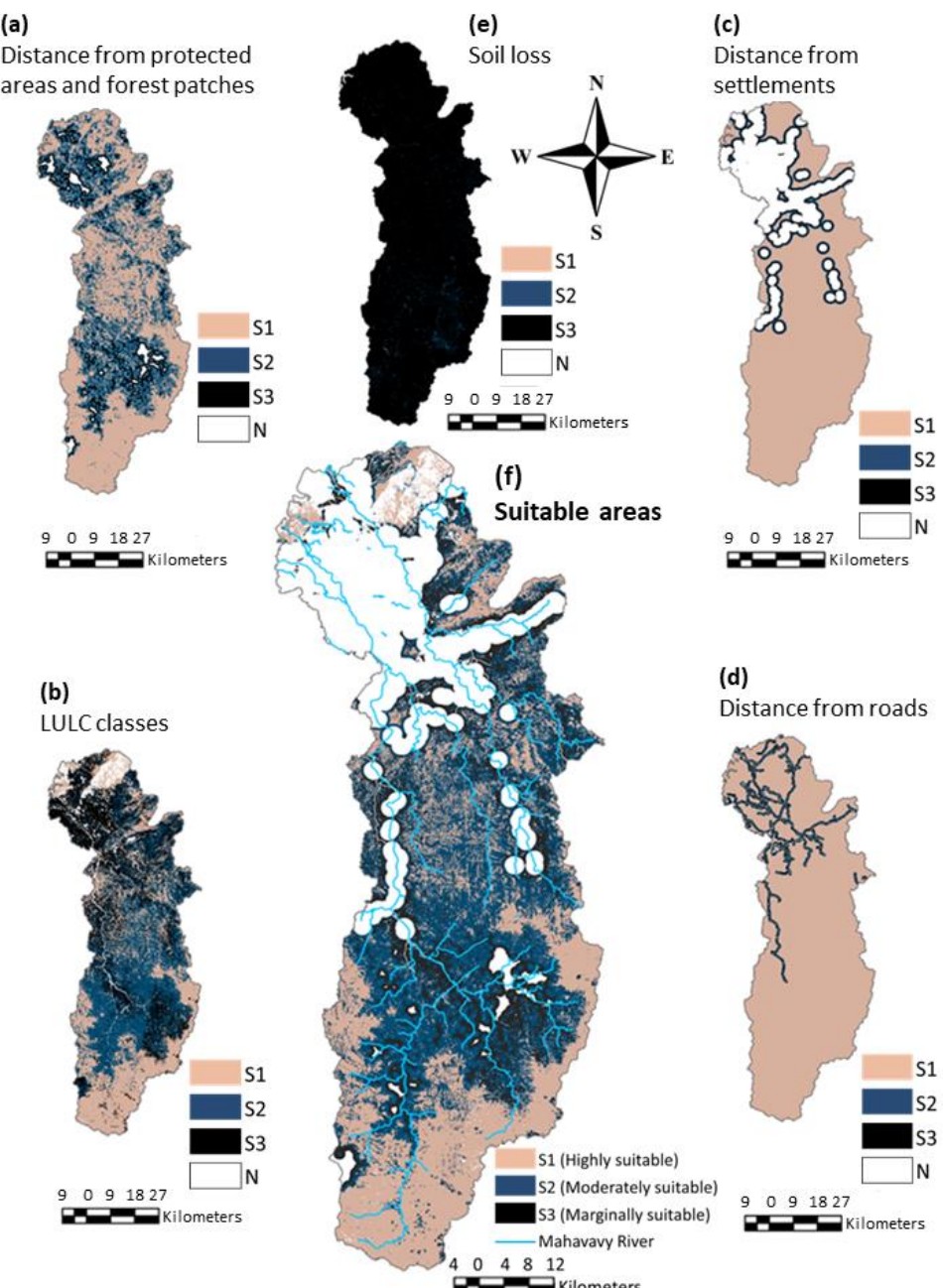

**Figure 7.** Reclassification of each criterion for land suitability analysis (**a**) distance from protected areas and forest patches, (**b**) LULC, (**c**) distance from settlements, (**d**) distance from roads, (**e**) soil loss, and (**f**) potential suitable areas for restoration in the Mahavavy watershed.

## 4. Discussion

The methodology used in this study is based on the FLR concept that shifts the attention from site to landscape restoration and guides our analysis through spatial multicriteria assessment. The GIS-based multicriteria analysis offers an effective tool to handle multi-objective problems such as the identification of restoration priority areas at the landscape scale [28]. From this concept, conflicting objectives that involve both ecological and socioeconomic were considered in our study. The basic assumption that guided the suggested approach is that a site could be given priority for restoration if it is sufficiently within or around forests and nature reserves (potential for increasing forest cover and protect ecologically important sites); not in close proximity to roads and settlements and in areas with high risk of soil erosion (feasibility of the restoration). The selection of criteria to represent

such factors was directly driven by literature review, computability, and data availability. From an ecological point of view, the criterion related to distance from protected areas and forest patches was chosen because increasing vegetation and forest cover in degraded land constitutes the primary restoration objective in the context of watershed protection and biodiversity conservation [59,63,64]. From a socioeconomic point of view, farther distance from settlements and roads were chosen because they minimize the cost of restoration. Furthermore, soil erosion was added to the socioeconomic criteria because restoration can protect and stabilize an erosion-prone land. Suitability maps were mostly generated from the use of distance calculation, which represents the vicinity to important ecological sites such as forests and other type of vegetation cover, and proximity to disturbance sources such as roads and settlements.

The new value used to standardize the classification in each suitability map assign a maximum value to areas in close proximity of forest patches and protected areas and minimize the value for areas near disturbance sources. As a result, high potential suitable areas are observed in close proximity of forest patches and protected areas, and low restoration feasibility in all areas that are easily accessible and thus subjected to exploitation. In addition, to prevent ambiguity related to human judgement in excluding areas not included in the highly suitable class, the use of moderate and marginally suitable classification partly compensates this problem.

As for how to prove the validity of the assessment outcomes of GIS-based multicriteria model, previous studies of land suitability assessment [6,28,33,48,65] found that that MCA&GIS models can be effectively used in the evaluation of land suitability. The results of the present study were cross-checked with the land suitable for FLR mentioned in the national document "National Strategy On Forest Landscape Restoration Furthermore, Green Infrastructure In Madagascar" to ensure the validity of the results using the method. The national document, developed by scientific experts and stakeholders, mentions degraded natural forests and degraded land as among the priority areas to restore in Madagascar [42], which was considered to be in the class of highly suitable in our results. Although it is important to include land cover change in the land suitability assessment [6], this evaluation was not included in the results shown here. However, the methodology we use can be applied to efficiently evaluate land suitability for FLR [28].

The GIS-based multicriteria approach used in this study for spatial decision-making has been applied by a number of applications related to specific prioritization studies [6,28,35,37,65,66]. Those studies showed that it is possible to evaluate maps of different criteria through multicriteria method and obtain suitability map for each criterion, and then combine all to generate the final suitability classification.

However, the interpretation of the results of this study should not lead to the conclusion that areas that have not been identified as high and moderately suitable are not important. In fact, our analysis only took into account five criteria considered important for identifying potential sites for watershed restoration and biodiversity conservation. These data are the most recent and complete available at the time of treatment.

## 5. Conclusions

This paper suggests the land suitability approach to identify priority areas for forest landscape restoration in tropical region in order to improve both ecosystem biodiversity and human well-being. The land suitability analysis is an application of the FLR framework in practice using GIS tool, that helps generate different options of suitable restoration areas. Areas that should be given priority for restoration are chosen based on sets of ecological and socioeconomic criteria. In this paper, we found that highly potential suitable areas for restoration are located in close proximity of forest patches and protected areas, and not suitable areas are those easily accessible and subjected to exploitation, such as near settlements and roads. Precisely, we have observed that with a total area of 514,789.2 ha, highly suitable areas for restoration account for 27.91% (143,680.14 ha), moderately suitable areas account for 30.91% (159,127.63 ha), and not suitable areas are 26.62% (137,031.23 ha)

in the watershed. However, it is important to mention that areas not included in the highly and moderate suitable classes are not necessarily considered having a low restoration feasibility. Those results may change depending on the level of precision we give by adding more criteria in the analysis. Despite the efforts, this study did not consider the deforestation dynamic,. Therefore, that is an important improvement that could be brought to this study in the future.

**Author Contributions:** Conceptualization, F.R.; methodology, F.R.; formal analysis, F.R.; investigation, F.R.; resources, R.A.W.; data curation, F.R.; writing—original draft preparation, F.R.; writing—review and editing, R.A.W.; visualization, F.R. and R.A.W.; supervision, R.A.W.; project administration, R.A.W.; funding acquisition, F.R. and R.A.W. All authors have read and agreed to the published version of the manuscript.

**Funding:** This research was funded by the Fulbright Fellowship Program and McIntire–Stennis Act of 1962 (P.L. 87-788), project number OHO00058-MS.

**Data Availability Statement:** Publicly available datasets were analyzed in this study. This data can be found here: [https://data.humdata.org]; [https://download.geofabrik.de/africa/madagascar.html]; [https://data.apps.fao.org/map/catalog/srv/eng/catalog.search#/metadata/446ed430-8383-11db-b9b2-000d939bc5d8]; [https://crudata.uea.ac.uk/cru/data/hrg/].

**Acknowledgments:** The authors would like to thank Emily Castellucci for her assistance and advice in the data computation and analysis.

**Conflicts of Interest:** The authors declare no conflict of interest. The funders had no role in the design of the study; in the collection, analyses, or interpretation of data; in the writing of the manuscript, or in the decision to publish the results.

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
