# Peer review of "Remote Sensing-Based Land Suitability Analysis for Forest Restoration in Madagascar"

_forests, doi:10.3390/f13101727_

Round 1

Reviewer 1 Report (Previous Reviewer 3)

The author has corrected some of the reviewers' suggestions. Some parts have not been taken correctly, such as the maps' specifications, the method of determining the weight and suitability range classes and their supporting references. More details are listed below (- = reviewer; + = author response):

- Line 57-59: This research is expected to produce a suitability map at a local scale. What map scale should be appropriate for a local scale? Doesn't an output map scale follow the smallest scale of the input maps? In this study, do the input maps support processing a local scale of FLR suitability map? All of this should be explained in this paper.

+ The research produces a suitability map at a landscape level, the scale of the watershed. We changed it to landscape level, instead of local level.

- Line 10: authors still use the local scale

- Table 1: add the map scale to the criteria column; for example, Land use/Land cover 1:100000

+ Map scales have been added to table 1.

- Map scales are inappropriate due to being too small, even though the analysis was carried out on one watershed. Maps generated from Landsat 8 interpretation should have a scale of at least 1:100000 or a maximum of 1:50000. How can the author produce a very small scale map for LULC and settlement maps? Table 1 mentions the use of Madagascar boundary map. In this study, the map of the Mahavavy watershed boundary should have been used.

- 301: describe the method used to determine the weight; include the literature, and display the weight value for each criterion in the results section.

+ This has been rewritten to clarify that weights were assigned to the groups ecological and socioeconomic, in which each group was considered equally important, with appropriate references. A table has been added to make further clarification of weighting in these two groups.

- I think the authors are wrong in determining the weight for each group; it should be ecological weight + socioeconomic weight = 1 (or 100%). Similarly, if added up, the weight of each criterion in one group should be = 1 (or 100%).

- The authors state that “each group was considered equally important” but the references referred to [6, 28] are irrelevant to this statement. Reference [6] does not specify an “equal weight”, but determines the weight based on the AHP analysis method and produces weights for Settlements 29%, LULC 29%, Elevation23%, Distance from roads12%, and Distance from rivers 7% (totally 100%). Reference [28] shows that the composition weights used for ecological (environment) and socioeconomic are not only “equal weight” (0.5 : 0.5), but they also tried to use a ratio of 0.7 : 0.3 and 0.3 : 0.7. The number and types of criteria used in reference [28] are also different from those used in this paper.

- Table 4: The method should describe how to determine the suitability class interval for each criterion. Was it determined at the discretion of the authors? Or are there certain references that are referred to because the values in Table 4 are not contained in references [1 & 23]. What are the justifications used by the author for each criterion? For example, in the distance from roads, why/how >500m is classified as S1, and <50m becomes N.

+ In table 4 we added the appropriate references as to how to determine the suitability class interval for each criterion. Justifications are added in the text of this section.

- I have checked all the references used by the author in Table 4. Those references do not match the reclassification in Table 4. The classification values ​​for criterion “Distance from protected areas and forest patches (m)” are not found in the reference [28]; The soil loss classified by the author does not match the soil loss class in the reference [56]. In this reference, soil loss (t/ha/y) is classified as <5 for the very low, 5-12 low, 12-50 Moderate, 50-100 severe, and >200 extremely severe categories. The author does not explain how to result the S1 > 500, S2 50 – 500, S3 0 – 50 (t ha-1 yr-1) classes; the values ​​of suitability range class for “Distance from settlements (m)” are not found in reference [7]; and the values ​​of suitability range class for “Distance from roads (m)” found in reference [6] are S1 < 10m, S2 10-23m, S3 23-45m, N >45m. The author cannot explain how to determine “Distance from roads (m)” so that S1 >500m, S2 250-500m, S3 50-250, N <50m are obtained.

Author Response

Comments and Suggestions for Authors

The author has corrected some of the reviewers' suggestions. Some parts have not been taken correctly, such as the maps' specifications, the method of determining the weight and suitability range classes and their supporting references. More details are listed below (- = reviewer; + = author response):

  • Line 57-59: This research is expected to produce a suitability map at a local scale. What map scale should be appropriate for a local scale? Doesn't an output map scale follow the smallest scale of the input maps? In this study, do the input maps support processing a local scale of FLR suitability map? All of this should be explained in this paper.

+ The research produces a suitability map at a landscape level, the scale of the watershed. We changed it to landscape level, instead of local level.

- Line 10: authors still use the local scale

à Yes, it’s landscape scale not local scale

  • Table 1: add the map scale to the criteria column; for example, Land use/Land cover 1:100000

+ Map scales have been added to table 1.

- Map scales are inappropriate due to being too small, even though the analysis was carried out on one watershed. Maps generated from Landsat 8 interpretation should have a scale of at least 1:100000 or a maximum of 1:50000. How can the author produce a very small scale map for LULC and settlement maps? Table 1 mentions the use of Madagascar boundary map. In this study, the map of the Mahavavy watershed boundary should have been used.

àWe added in the table the resolution of the Landsat satellite imagery that we used to produce the LULC and settlement maps. It is a 30 m resolution image.

  • 301: describe the method used to determine the weight; include the literature, and display the weight value for each criterion in the results section.

+ This has been rewritten to clarify that weights were assigned to the groups ecological and socioeconomic, in which each group was considered equally important, with appropriate references. A table has been added to make further clarification of weighting in these two groups.

- I think the authors are wrong in determining the weight for each group; it should be ecological weight + socioeconomic weight = 1 (or 100%). Similarly, if added up, the weight of each criterion in one group should be = 1 (or 100%).

à We made a mistake in our previous version. The weight for each group is 0.50, which is equal to 1 when added.

- The authors state that “each group was considered equally important” but the references referred to [6, 28] are irrelevant to this statement. Reference [6] does not specify an “equal weight”, but determines the weight based on the AHP analysis method and produces weights for Settlements 29%, LULC 29%, Elevation23%, Distance from roads12%, and Distance from rivers 7% (totally 100%). Reference [28] shows that the composition weights used for ecological (environment) and socioeconomic are not only “equal weight” (0.5 : 0.5), but they also tried to use a ratio of 0.7 : 0.3 and 0.3 : 0.7. The number and types of criteria used in reference [28] are also different from those used in this paper.

à We only use the reference [28] as a guide for our weight calculation, particularly to perform the equal weight (0.5 : 0.5) in their method, although the number and types of criteria are different than the ours. However, we did not use the ration 0.7:0.3 or 0.3 : 0.7.

  • Table 4: The method should describe how to determine the suitability class interval for each criterion. Was it determined at the discretion of the authors? Or are there certain references that are referred to because the values in Table 4 are not contained in references [1 & 23]. What are the justifications used by the author for each criterion? For example, in the distance from roads, why/how >500m is classified as S1, and <50m becomes N.

+ In table 4 we added the appropriate references as to how to determine the suitability class interval for each criterion. Justifications are added in the text of this section.

- I have checked all the references used by the author in Table 4. Those references do not match the reclassification in Table 4. The classification values ​​for criterion “Distance from protected areas and forest patches (m)” are not found in the reference [28]; The soil loss classified by the author does not match the soil loss class in the reference [56]. In this reference, soil loss (t/ha/y) is classified as <5 for the very low, 5-12 low, 12-50 Moderate, 50-100 severe, and >200 extremely severe categories. The author does not explain how to result the S1 > 500, S2 50 – 500, S3 0 – 50 (t ha-1 yr-1) classes; the values ​​of suitability range class for “Distance from settlements (m)” are not found in reference [7]; and the values ​​of suitability range class for “Distance from roads (m)” found in reference [6] are S1 < 10m, S2 10-23m, S3 23-45m, N >45m. The author cannot explain how to determine “Distance from roads (m)” so that S1 >500m, S2 250-500m, S3 50-250, N <50m are obtained.

à We have already provided more paragraphs that explain the determination of the ranking system of each criteria. We also added more literature.

à For some references that we listed in our study, we only used them as a guide for ranking system, but did not copy-paste the values that they used in their studies due to the differences observed in different regions.

Reviewer 2 Report (Previous Reviewer 2)

Dear authors.

Thank you very much for your provided responses to my comments. I think that you significantly improved your study, comparing to the previous version. However, I have to recommend some minor comments. You did not address properly some of my comments. In lines 47-52 (previous version) you did not add the proposed literature. Please, respond accordingly and add the literature. Also, in my comments in lines 305-314 (previous version, 3.1. Soil Quantification and Mapping) you explained that you did not find similar research to compare the erosion values. However, in the text you did not say/comment something. You should add a phrase (as limitation) and say that you can not validate/compare the erosion values, because of the absence of similar studies in the region. In lines 288-289 (in the latest version) you say: "The amount of soil loss is strongly dependent on rainfall, runoff, type of soil, vegetation and topographic characteristics [51]". I agree with you. However, there also some other biotic/abiotic factors that significantly impact the soil loss. I strongly propose to modify this phrase like that: "The amount of soil loss is strongly dependent on rainfall, runoff, type of soil, vegetation, biotic/abiotic disturbances and topographic characteristics [51]". And here, with the literature [51], you should add the two following studies that examined the biotic/abiotic disturbances impact on soil loss (https://doi.org/10.3390/land11060911 and https://doi.org/10.1016/j.scitotenv.2021.150106). Finally, you made a lot of changes in the text. So, you have to be careful when preparing the final version. 

Author Response

REVIEWER 2

Thank you very much for your provided responses to my comments. I think that you significantly improved your study, comparing to the previous version.

  • However, I have to recommend some minor comments. You did not address properly some of my comments. In lines 47-52 (previous version) you did not add the proposed literature. Please, respond accordingly and add the literature.

à We already added the proposed literature

  • Also, in my comments in lines 305-314 (previous version, 3.1. Soil Quantification and Mapping) you explained that you did not find similar research to compare the erosion values. However, in the text you did not say/comment something. You should add a phrase (as limitation) and say that you can not validate/compare the erosion values, because of the absence of similar studies in the region.

à We added: “As the limitation of this study, the estimation of soil loss quantification and mapping as well as the comparison of the erosion values were based on the soil loss potential of the Abay river basin in Ethiopia, Africa (Duguma, 2022), due to the lack of similar studies in the study region.

  • In lines 288-289 (in the latest version) you say: "The amount of soil loss is strongly dependent on rainfall, runoff, type of soil, vegetation and topographic characteristics [51]". I agree with you. However, there also some other biotic/abiotic factors that significantly impact the soil loss. I strongly propose to modify this phrase like that: "The amount of soil loss is strongly dependent on rainfall, runoff, type of soil, vegetation, biotic/abiotic disturbances and topographic characteristics [51]". And here, with the literature [51], you should add the two following studies that examined the biotic/abiotic disturbances impact on soil loss (https://doi.org/10.3390/land11060911 and https://doi.org/10.1016/j.scitotenv.2021.150106).

à We added the proposed phrase and literature

  •  
  • Finally, you made a lot of changes in the text. So, you have to be careful when preparing the final version.

Round 2

Reviewer 1 Report (Previous Reviewer 3)

The authors have provide coherent feedback on all corrections and suggestions.

This manuscript is a resubmission of an earlier submission. The following is a list of the peer review reports and author responses from that submission.

Round 1

Reviewer 1 Report

Dear Authors,

Thank you for the interesting submission to Forests. The paper you have provided is in fact a suitability study with a focus on forest restoration. It is an interesting approach towards these types of analyses, but I need to see more clearly expressed the novelty in this work. Could you relate somehow the potential areas for forest restoration with LU change which should take place. What is the estimated cost for land use conversion and what will be the ecosystem services improvement if the scenario outlined is realized. At the point where this work finishes it could be of help for policymakers and practitioners, but are these targets connected somehow with international engagements of Madagascar? There are many other smaller issues which I will address on the next review round if you properly address these big issues.

Kind regards,

Reviewer

Reviewer 2 Report

Dear Editor.

I have finished my review on the proposed paper “Remote sensing-based land suitability analysis for forest restoration in Madagascar”, forests-1885688-peer-review-v1.

Summary of the manuscript:

In the proposed paper, the authors’ goal is to identify priority areas for forest landscape restoration at a local scale using a geospatial land suitability analysis approach. They used multiple 5 criteria in combination with GIS techniques and multiple decision making, in order to locate the appropriate location for reforestation. They found that there are very large areas that are suitable for forest landscape restoration (27.9% highly suitable areas for forest landscape restoration and 30.9% moderately suitable).

General review:

1. Generally, the manuscript presents an interesting topic and the specific research seems to include some significant points for the research community of this field. The adding value of this paper is that the research area is in Madagascar, which a very sensitive, unique and threatened ecosystem.

2. The proposed paper is very well written with very good use of English language. Except some very minor grammatical mistakes and word errors. The authors should check again the paper to correct these minor mistakes.

3. The proposed paper is very well structured. It begins with the Introduction with some limited references that helps the reader to get into the subject. In Introduction there is an effort to provide previous studies with similar scientific content, which took place in the research area and in other countries. At the end of Introduction, authors clearly state the goals of the research. However, the authors didn’t manage to provide adequate number of relative studies that recently published, which deal with the same subject and used almost the same methodology. Below, I gave some examples.

4. The methodology is generally interesting, and explained, so other researchers could easily repeat it. However, some parts need to be clearer.  See below specific comments.

5. The results are OK.

6. The quality of the work in Discussion is not adequate. See below specific comments.

7. Conclusions are appropriate for this paper.

Additional points for revision:

In my opinion, the proposed paper could be characterized as a good research work, complies with aims of Forests. 

INTRODUCTION: This part of the study is very weak. More literature should be added from previous studies that deal with Multi-Criteria Evaluation and GIS. You used only 28 references. For the specific subject less than 40 references is not adequate. You should find these kinds of studies, add them in Introduction and define what is new/novel in your research. Below, I give some examples.

Lines 27-29: Here it is a statement, please add literature.  

Lines 47-52: Here, you say a few words about the GIS and land suitability assessment using multiple criteria. However, you failed to represent an adequate literature review (state-of-the-art) of previous studies and highlight the novelty of your research. You should add a separate paragraph. I proposed to add the following studies and take into account the reference lists of these studies. These studies used multiple criteria and GIS for reforestation purposes (Tzioutzios et al 2020 https://doi.org/10.3390/ijgi9120725, Van der Horst and Gimona https://doi.org/10.1016/j.biocon.2004.11.020, Eastman 1999).

Lines 133-134: Here, you used only one reference [15]. You should add more from the above proposed and other that you will find.

Lines 157-158: “…..equation (1) was driven by literature review and the need to be spatially represented.”. Where is this literature. You should the respective references.

Lines 305-314, 3.1. Soil Quantification and Mapping: Here, are the results of RUSLE and the perspective categorization, severe risk of erosion, low to moderate erosion and insignificant risk of erosion. The values that you erosion values that you calculated were not compared with studies from similar tropical regions. You should find some studies with measured erosion values and compare your results, in order to (some kind) validate the results of RUSLE.

Figure 6 and 7 and generally in the text: You indicate in the results that the dense forest area is highly suitable for reforestation. In methodology (table 4) you set as “Highly suitable (S1)” both the Forest (dense and degraded). However, a simple question can arise here: “how it is possible to set as Highly suitable (S1) areas for reforestation, the areas that are already dense forest??? What will be reforested in an already dense forest???”. Maybe you should exclude from your analysis the dense forest areas.

DISCUSSION: The Discussion is weak. Discussion within the context of comparing the results of the paper with other studies, in not exists. I searched the paper, but I found very few references. You should compare your results with previously published studies.

References

Eastman, R. 1999. Multi-criteria evaluation in GIS. In Geographical Information Systems; Longley, P.A., Goodchild, M.F., Maguire, D.J., Rhind, D.W., Eds.; Whiley: New York, NY, USA, 1999; Chapter 35; pp. 493–502.

Van der Horst, D.; Gimona, A. Where new farm woodlands support biodiversity action plans: A spatial multi-criteria analysis. Biol. Conserv. 2005, 123, 421–432.

Tzioutzios, C.; et al. A. Multi-Criteria Evaluation (MCE) Method for the Management of Woodland Plantations in Floodplain Areas. ISPRS Int. J. Geo-Inf. 2020, 9, 725.

Reviewer 3 Report

This manuscript is in line with the scope of FORESTS, and its contents are useful to support the speeding up Forest Landscape Restoration in Madagascar. Some improvements must be made, especially in the method section and related references.

Line 57-59: This research is expected to produce a suitability map at a local scale. What map scale should be appropriate for a local scale? Doesn't an output map scale follow the smallest scale of the input maps? In this study, do the input maps support processing a local scale of FLR suitability map? All of this should be explained in this paper.

Line 80: C or F degree?

Table 1: add the map scale to the criteria column; for example, Land use/Land cover 1:100000

The link in Table 1 (4) should be: https://download.geofabrik.de/africa/madagascar.html

228: Authors state that the settlement map is available. Is the source of information for the distance from the settlement the same as the road? In table 1, the settlement map comes from Landsat 8 and Google Earth, so must it go through processing?

231: Cite references related to image processing used in this study.

237-238: what is the LULC class reference? Is it following the FAO land cover classification system?

246-248: add references/literature to support the methods of calculating the sample size, Producer's Accuracy (PA), User's Accuracy (UA) and Overall Accuracy (OA).

260: add "original" source/literature for this equation.

301: describe the method used to determine the weight; include the literature, and display the weight value for each criterion in the results section.

Table 4: The method should describe how to determine the suitability class interval for each criterion. Was it determined at the discretion of the authors? Or are there certain references that are referred to because the values in Table 4 are not contained in references [1 & 23]. What are the justifications used by the author for each criterion? For example, in the distance from roads, why/how >500m is classified as S1, and <50m becomes N.